# *Atelerix algirus,* the North African Hedgehog: Suitable Wild Host for Infected Ticks and Fleas and Reservoir of Vector-Borne Pathogens in Tunisia

**DOI:** 10.3390/pathogens10080953

**Published:** 2021-07-29

**Authors:** Ghofrane Balti, Clemence Galon, Moufida Derghal, Hejer Souguir, Souheila Guerbouj, Adel Rhim, Jomâa Chemkhi, Ikram Guizani, Ali Bouattour, Sara Moutailler, Youmna M’ghirbi

**Affiliations:** 1Laboratory of Viruses, Vectors and Hosts, LR20IPT02, Institut Pasteur de Tunis, Université Tunis El Manar, 13, Place Pasteur, Tunis 1002, Tunisia; b.ghofraane@gmail.com (G.B.); adel.rhim@yahoo.fr (A.R.); ali.bouattour@pasteur.tn (A.B.); 2Unité Mixte de Recherche de Biologie Moléculaire et d’Immunologie Parasitaires, Animal Health Laboratory, Agence Nationale de sécurité Sanitaire de l’Alimentation, de l’Environnement et du Travail, National Veterinary School of Alfort, Paris-Est University, Maisons-Alfort, 94700 Paris, France; clemence.galon@anses.fr; 3Laboratory of Molecular Epidemiology and Experimental Pathology, LR16IPT04, Institut Pasteur de Tunis, Université Tunis El Manar, Tunis 1002, Tunisia; dgl.moufida@gmail.com (M.D.); souguirhejer@gmail.com (H.S.); souheila.guerbouj@gmail.com (S.G.); chemkhijomaa@gmail.com (J.C.); ikram.guizani@pasteur.tn (I.G.)

**Keywords:** pathogens, hedgehogs, vectors, zoonotic diseases, microfluidic real-time PCR

## Abstract

Small wild mammals are an important element in the emergence and transmission of vector-borne pathogens (VBPs). Among these species, hedgehogs have been found to be a reservoir of VBPs and host of arthropod vectors. Surveillance of VBPs in wildlife and their arthropods are crucial in a one health context. We conducted an exploratory study to screen *Atelerix algirus* hedgehogs and their infesting ticks and fleas for VBPs using a high throughput microfluidic real-time PCR system. Tested biopsies from hedgehogs were found to be naturally infected by *Theileria youngi*, *Hepatozoon* sp., *Ehrlichia ewingii*, *Coxiella burnetii*, and *Candidatus* Ehrlichia shimanensis. Similarly, *Haemaphysalis erinacei* and *Rhipicephalus sanguineus* tick species were infected by *Ehrlichia ewingii*, *Rickettsia* spp., *Rickettsia massiliae*, *Borrelia* sp., *Coxiella burnetii*, *Rickettsia lusitaniae* and *Anaplasma* sp. *Archaeopsylla erinacei* fleas were infected by *Rickettsia asembonensis*, *Coxiella burnetii*, and *Rickettsia massiliae.* Co-infections by two and three pathogens were detected in hedgehogs and infesting ticks and fleas. The microfluidic real-time PCR system enabled us not only to detect new and unexpected pathogens, but also to identify co-infections in hedgehogs, ticks, and fleas. We suggest that hedgehogs may play a reservoir role for VBPs in Tunisia and contribute to maintaining enzootic pathogen cycles via arthropod vectors.

## 1. Introduction

Wild fauna has always been considered to play a fundamental role in the emergence and re-emergence of zoonotic diseases in nature. In fact, most emerging zoonotic pathogens are of wild animal origin [1]. Major drivers of zoonotic disease emergence and spillover include human activities such as urbanization and landscape modification, which disrupt the ecosystems of wild mammal hosts [2,3,4]. This is true especially for vector-borne diseases (VBDs) that have multi-element transmission cycles and that could be directly or indirectly affected by ecosystem disruptions [5]. In such transmission cycles, wild mammal hosts play a prominent role in the amplification and/or transmission of pathogens; they are also suitable hosts for hematophagous arthropods [6,7,8].

VBDs represent a considerable challenge in a one health perspective in view of transmission and pathogen diversity, and human and animal exposure risk and mortality. Furthermore, the discovery and emergence of new pathogens, due in most cases to the invasion of new habitats by vector species and wild reservoir hosts, highlight the need for an intensified surveillance and well-developed investigations [9]. Among this wildlife, hedgehogs (Eulipotyphla: Erinaceidae) could play an important role in the emergence of zoonotic vector-borne pathogens. Hedgehogs, small insectivorous wild-living mammals with nocturnal habits, are distributed throughout most of the temperate and tropical areas of Europe, Asia and Africa, and in New Zealand by introduction [10,11]. Different species are reported: *Erinaceus europaeus* (Linnaeus, 1758) is commonly reported in European countries; however, *Atelerix algirus* (Lereboullet, 1842) is native to the northern regions of Africa from Morocco to Libya, and to the Balearic and Canary islands [10,12]. In Tunisia, *A. algirus* has recently been reported to live in sympatry with the desert hedgehog *Paraechinus aethiopicus* [10,13].

These animals are highly adaptable denizens of urban and suburban areas [13,14]. They are commonly infested with different ectoparasites, mainly hard ticks (Ixodidae) and fleas (Siphonaptera) [15,16,17,18,19,20] of particular medical and veterinary interest. This can increase the risk of direct exposure of humans and companion animals to arthropods [21].

The European hedgehog, *Erinaceus europaeus*, and its infesting ticks have been found to be infected with *Borrelia burgdorferi* (*s.l*.) genospecies [22,23,24], *Anaplasma phagocytophilum* [25,26], *Rickettsia helvetica* [27] and tick-borne encephalitis virus (TBEV) [28]. Moreover, *Neoehrlichia mikurensis* and *Anaplasma phagocytophilum* were detected in Northern white-breasted hedgehog (*Erinaceus roumanicus*) tissue samples in Hungary [29]. Recently, *Coxiella burnetii*, the causative agent of Q fever, was detected in *Erinaceus amurensis* hedgehogs in China [30]. These studies suggest that hedgehogs may serve as reservoir hosts for several zoonotic vector-borne pathogens (VBPs) and could contribute to their enzootic cycles in nature.

In contrast to European hedgehog species, few studies have been performed exploring VBPs that may occur in the North African hedgehog, *Atelerix algirus*. Investigations conducted in Algeria reported infection of *A. algirus* with *Bartonella tribocorum* and *B. elizabethae* [31]; in addition, *Rickettsia felis* and *Rickettsia massiliae* have been detected in fleas and ticks infesting this hedgehog species, respectively [19,32]. Furthermore, *A. algirus* was proven to be a potential reservoir for *Leishmania major* and *Leishmania infantum* in Algeria and Tunisia [33,34,35].

The implication of hedgehogs in the transmission and maintenance of several emerging etiological agents of public health concern in Tunisia has not been elucidated to date. As a result, using large-scale high-throughput screening, we aimed to explore whether hedgehogs in Tunisia contribute to the enzootic cycle of vector-borne bacteria and protozoa, and to shed more light on the mechanisms of transmission cycles involving hedgehogs, and their infesting tick and flea species. To the best of our knowledge, this is the first large-scale high-throughput screening investigation of vector-borne bacteria and protozoa in *Atelerix algirus* hedgehogs and their ectoparasites in Tunisia.

## 2. Results

### 2.1. Investigated Hedgehogs and Infesting Arthropods

Based on external morphological criteria, all captured hedgehogs (*n* = 20) were identified as *Atelerix algirus* (Table 1). A total of 105 tissue samples were obtained after hedgehog dissection (20 spleens, 20 livers, 17 kidney, 18 hearts, 10 lymph nodes, 12 blood and 8 bone marrow fluids).

A total of 110 ticks and 92 fleas were collected from nine hedgehogs (ED1, EZ4, EB1–EB6, and EG1). The remaining hedgehogs (EA1–EA7; EZ1–EZ3) were not examined for the presence of ectoparasites (Table 1). The majority of ticks were semi-engorged and identified as *Haemaphysalis erinacei* (*n* = 92), followed by *Rhipicephalus sanguineus* (*n* = 15), and *Hyalomma aegyptium* (*n* = 1); only two *Ixodes* spp. were fully engorged. Moreover, the *Archaeopsylla erinacei* flea was the most common species collected from hedgehogs (*n* = 91), in addition to one specimen of *Ctenocephalides felis*.

Among the 10 hedgehogs examined for arthropod infestation, nine were infested with ticks and fleas. Among them, only one was infested solely by ticks. The number of ticks and fleas infesting hedgehogs ranged from 0 to 55 (mean = 11; 10% [95% Confidence interval: 8.2–13.8]) and 0 to 51 (mean = 9.2; 10% [95% Confidence interval: 6.15–12.3]), respectively. Interestingly, one hedgehog (EB1) captured from the Bizerte region was heavily infested and carried almost half of the total collected ticks (55/110; 50%) and fleas (51/92; 55.4%) (Figure 1).

### 2.2. Vector-Borne Pathogen Detection in Hedgehog Biopsies and Infesting Arthropods

A total of 105 biopsies sampled on 20 *Atelerix algirus* hedgehogs, 110 ticks, and 92 fleas were screened individually for the presence of vector-borne bacteria (*Rickettsia*, *Anaplasma, Ehrlichia, Bartonella, Borrelia, Coxiella,* and *Francisella*) and protozoa (*Babesia*, *Theileria*, and *Hepatozoon*) using microfluidic real-time PCR on the BioMark^TM^ system. The number of positive samples revealed by microfluidic real-time PCR and the corresponding infection rates (IRs) are summarized in Table 2.

To confirm the results obtained by the microfluidic real-time PCR system, conventional PCRs or nested PCRs followed by sequencing were performed on the positive samples. Similarities of the obtained sequences with the available reference sequences in GenBank (NCBI) are presented in Table 3.

#### 2.2.1. Vector-Borne Pathogen Detection in Atelerix Algirus Hedgehogs

Using the microfluidic real-time PCR system, nine *A. algirus* (9/20; 45%) were positive for *Ehrlichia* spp. (Table 2). Among 105 analyzed biopsies, 14 were positive for *Ehrlichia* spp. (13.3%). Among them, one had three infected organs (heart, spleen, and liver); and three hedgehogs had two infected organs: blood and either liver, heart or bone-marrow fluid. The remaining *A. algirus* (five out of nine) showed infection in either blood, kidney, spleen or bone marrow fluid.

To further confirm the occurrence of *Ehrlichia* species in hedgehog biopsies, we were able to obtain four sequences by amplifying a 16S rRNA gene fragment by nested PCR (Table 3). We revealed two different genotypes: (i) three sequences obtained from the liver, heart, and kidney (accession numbers MW508468, MW508474, and MW508475) displayed 99.15–99.33% identity with *Candidatus* Ehrlichia shimanensis (accession number AB074459), (ii) one sequence obtained from a blood sample (accession number MW508473) showed 99.7% identity with *Ehrlichia ewingii* (accession number MN148616). The *E. ewingii* and *Candidatus* Ehrlichia shimanensis sequences identified in this study were in the same cluster as several *E. ewingii* and *Candidatus* Ehrlichia shimanensis sequences available in GenBank, respectively (Figure 2).

*Theileria* spp. DNA was detected in eight *A. algirus* (8/20, 40%). Fourteen of 105 (13.3%) biopsies from these hedgehogs were found to be *Theileria*-positive. Among these hedgehogs, three had two infected organs: liver and either heart, kidney or blood; while one specimen had an infection in three different samples (liver, blood, and lymph node). The remaining hedgehogs (*n* = 4) were infected in solely one organ: in the blood (*n* = 3), and in the heart (*n* = 1). Using microfluidic real-time PCRs, no signal was obtained for the two targeted *Theileria* species (*T. velifera* and *T. mutans*) suggesting infection by another *Theileria* species. The partial sequences of the 18S rRNA gene obtained (accession numbers MW508493, MW508494, and MW508496) were 98.9–99.7% similar to *Theileria youngi* (accession number AF245279) (Table 2 and Table 3). *Theileria youngi* sequences obtained in this study are within the same cluster as the reference sequence of *T. youngi* in GenBank.

Furthermore, the liver of one hedgehog (1/20, 5%) was positive for *Hepatozoon* spp. (Table 2). Amplification of 18S rRNA using nested PCR and sequencing of the PCR product revealed a sequence (accession number MW508490) 100% identical to uncharacterized *Hepatozoon* species (accession number KU680466) (Table 2). Our sequence clustered with several uncharacterized *Hepatozoon* sp. (Figure 3).

*Coxiella burnetii* DNA was detected in two *A. algirus* hedgehogs (2/20, 10%, Table 2): one hedgehog was infected in three different biopsies (spleen, liver, and lymph node), while the other hedgehog had a blood infection. To confirm the microfluidic real-time PCR results, we amplified and sequenced a 16S rRNA gene fragment. Only one sequence was successful (accession number MW508461) which showed 100% identity (Table 3) with the *Coxiella burnetii* strain SFA062 from humans (accession number MN540441). 

Lastly, two hedgehogs (2/20, 10%), were positive for *Rickettsia* spp. (Table 2). One was infected in three different biopsies (spleen, heart, and liver), while the other was infected in the liver. Unfortunately, neither conventional PCR targeting gltA nor nested PCR targeting ompB succeeded in confirming the microfluidic real-time PCR results.

#### 2.2.2. Vector-Borne Pathogen Detection in Ticks

The most common pathogen detected in *Haemaphysalis erinacei* and *Rhipicephalus sanguineus* ticks collected from hedgehogs was *Coxiella burnetii*, with an infection rate reaching 80.4% (74/92) and 86.6% (13/15), respectively (Table 2). In addition, *C. burnetii* was also detected in one of the two fully engorged *Ixodes* spp. and in the only *Hyalomma aegyptium* tested. The presence of *C. burnetii* was confirmed by nested PCR targeting the 16S rRNA gene in six randomly chosen positive *Hae. erinacei* ticks. Obtained sequences (accession numbers MW508462-MW508467) were 99–100% similar to *C. burnetii* strain CB-30 and *C. burnetii* strain SFA062 (accession numbers LC46497 and MN540441) (Table 3). Unfortunately, none of the randomly chosen *Rh. sanguineus*
*Coxiella burnetii*-positive samples (*n* = 3) were successfully amplified by nested PCR to confirm corresponding microfluidic results.

*Rickettsia* spp. were detected in 40.2% of tested *Hae. erinacei* ticks (37/92) and in 86.6% of *Rh. sanguineus* (13/15) (Table 2). We did not succeed in amplifying the gltA or ompB genes to further identify the *Rickettsia* species in *Hae. erinacei* by targeted PCR. However, 8 of 15 *Rh. sanguineus* ticks were found to be infected by *Rickettsia massiliae* (Table 2). For confirmation, initial attempts to amplify the gltA gene led to successful amplification of 2/8 *Rickettsia* DNAs (Table 3). One of the sequences (accession number MW508482) showed 100% identity with *R. massiliae* (accession number DQ503428). The second gltA sequence (accession number MW508481) showed 100% identity with *R. lusitaniae* (accession number KC428021). To further confirm the results of the remaining six positive *Rh. sanguineus*, we targeted the ompB gene. Corresponding sequences (accession numbers MW508484-MW508489) displayed 99–100% identity with *R. massiliae* (accession number MK761227) (Table 3). *Rickettsia massiliae* sequences obtained in this study clustered with GenBank published *R. massiliae* as shown in the phylogenetic tree (Figure 4). Likewise, the *R. lusitaniae* sequence found in our samples formed a cluster with the deposited *R. lusitaniae* ones (Figure 5).

*Ehrlichia* spp. was detected in 26% of tested *Hae. erinacei* (24/92), while no *Rh. sanguineus* were positive (Table 2). To confirm this result and to identify *Ehrlichia* species, we successfully amplified *Ehrlichia* DNA in two *Hae. erinacei* by conventional PCR targeting the 16S rRNA gene (Table 3). Sequences (accession numbers MW508471-MW508472) showed 99.8% and 100% identity with *Ehrlichia ewingii* strain Hubei CW46 (accession number MN148616) (Table 2). Our *Ehrlichia ewingii* sequences clustered within several *E. ewingii* sequences deposited in GenBank (Figure 2).

Moreover, one *Hae. erinacei* and one *Rh. sanguineus* were found to be infected by *Anaplasma* spp. (Table 2). This result was confirmed by conventional PCR targeting the 16S rRNA gene and the corresponding sequence obtained from *Rh. sanguineus* (accession number MW508491) showed 99.7% identity with an uncharacterized *Anaplasma* sp. BL102-7 (accession number KJ410249). Likewise, the sequence obtained from *Hae. erinacei* (accession number MW508469) showed 99.7% identity with *Ehrlichia ewingii* strain Hubei CW46 (accession number MN148616) (Table 3). *Anaplasma* spp. identified in this study clustered with *A. phagocytophilum*, *Candidatus* Anaplasma boleense, and several uncharacterized *Anaplasma* species (Figure 6).

One *Rh. sanguineus* was positive for *Borrelia* spp. but not *Borrelia burgdorferi* s. l. species (the Lyme disease agent), nor the *Borrelia* relapsing fever group primers/probes sets (Table 2). For confirmation, nested PCR targeting the flaB gene was performed and followed by PCR products sequencing. The sequence (accession number MW508492) obtained from this tick was 99.70% similar to *Borrelia* sp. clone Ir-Maz190 (accession number MN958351) (Table 3). Phylogenetic analysis showed that *Borrelia* sp. obtained in this study formed a cluster with some *Borrelia* species belonging to the relapsing fever group, such as *Borrelia lonestari*, *B. theileri*, and uncharacterized *Borrelia* species (Figure 7).

Three *Hae. erinacei* (3.3%, 3/92) and one *Rh. sanguineus* (6.7%, 1/15) (Table 2) were positive for *Bartonella* spp. Unfortunately, sequencing attempts of the confirmatory PCR products failed.

#### 2.2.3. Vector-Borne Pathogen Detection in Fleas

*Rickettsia* spp. DNA was detected in 82.4% of *Archaeopsylla erinacei* (75/91) and in one *Ctenocephalides felis* (1/1) (Table 2). However, none of the targeted *Rickettsia* species of the BioMark^TM^ system gave a positive signal, suggesting the presence of unexpected *Rickettsia* species in the tested hedgehog fleas. To identify the *Rickettsia* species, we amplified the gltA and ompB genes by nested PCR, followed by sequencing. Four identical gltA sequences (accession numbers MW508476-MW508479) obtained from four *Archaeopsylla erinacei* were 100% identical to *Rickettsia asembonensis* (accession number MN186290) (Table 3). Furthermore, the two partial sequences of the ompB gene fragment from two *A. erinacei*, which infested two different hedgehogs, were 100% similar (accession number MW508480) to *R. asembonensis* (accession number MK923741) and 99.3% to *R. massiliae* (accession number AF123714), respectively. The phylogenetic tree of *R. asembonensis* and *R. massiliae* sequences in this study showed that the sequences were in the same cluster with corresponding reference sequences (Figure 4 and Figure 5). The *R. massiliae* sequence amplified in fleas was identical to *R. massiliae* identified in ticks.

*Coxiella burnetii* DNA was directly detected by the BioMark^TM^ system in 34.1% of *A. erinacei* (31/91) and one *C. felis* (1/1) (Table 2). Nested PCR targeting the 16S rRNA gene followed by sequencing of one randomly chosen *A. erinacei* confirmed the microfluidic real-time PCR results. The corresponding sequence (accession number MW508460) was similar to *C. burnetii* strain SFA062 (accession number MN540441) (Table 2).

Two *A. erinacei* fleas (2.2%, 2/91) were positive for *Bartonella* spp. by microfluidic-real-time PCR. Unfortunately, conventional PCR targeting either gltA or ITS followed by sequencing did not succeed in confirming the microfluidic PCR results.

#### 2.2.4. Co-Infection in Hedgehogs, Ticks and Fleas 

Among the 20 tested *Atelerix algirus*, seven (7/20, 35%) revealed co-infection by at least two pathogens. Double infections were observed in four hedgehogs (20%), including infection by *Theileria youngi* and *Ehrlichia* spp., while one subject was co-infected by *Rickettsia* spp. and *Hepatozoon* sp. Interestingly, a triple infection with *C. burnetii*, *T. youngi*, and *Ehrlichia* spp. was observed in two hedgehogs (10%).

Additionally, co-infections were revealed in *Haemaphysalis erinacei* and in *Rhipicephalus sanguineus* ticks. Almost 45.6% of tested *Hae. erinacei* (42/92) and 80% of *Rh. sanguineus* (12/15) were naturally infected by at least two pathogens. In addition, 21% of *Hae. erinacei* (19/92) were co-infected by three pathogens: 18 *Hae. erinacei* were co-infected by *C. burnetii*, *Rickettsia* spp., and *Ehrlichia* spp., while one specimen was infected by *C. burnetii*, *Rickettsia* spp., and *Bartonella* spp. The double infection observed in *Hae. erinacei* was either with *C. burnetii* and *Rickettsia* spp. (12/92; 13%) or with *C. burnetii* and *Ehrlichia* spp. (10/92; 10.8%).

A total of 10 *Rh. sanguineus* (66.6%) were co-infected with *C. burnetii* and *Rickettsia massiliae*, while two *Rh. sanguineus* (2/15, 13.3%) presented a triple infection with *C. burnetii*, *R. massiliae*, and *Borrelia* sp. or *Anaplasma* sp.

Fleas also presented multiple infections, 30.8% (28/91) of *A. erinacei* were co-infected with at least two pathogens. Double infections were observed in 29.6% (27/91) of *A. erinacei*, including *C. burnetii* and *R. asembonensis* (*n* = 26), and *Bartonella* spp. and *Rickettsia* spp. (*n* = 1). Triple infection with *C. burnetii*, *Bartonella* spp., and *Rickettsia* spp. was detected in one *A. erinacei*. Moreover, *Ctenocephalides felis* (1/1) also presented a co-infection with *C. burnetii* and *Rickettsia* spp.

## 3. Discussion

Wild animals such as hedgehogs can serve a reservoir role for several zoonotic pathogens and thus represent a major public health concern, affecting all continents. In our study, all captured hedgehogs in North Tunisia were identified as *Atelerix algirus*. This small mammal is considered the main hedgehog species in Tunisia. It is an endemic animal of the Maghreb region (Morocco, Algeria, Tunisia and Libya) where it colonizes a wide variety of biotopes. In contrast, the second most common hedgehog species encountered in Tunisia is *Paraechinus aethiopicus*; it is recorded in the center and the south, with a specialization in the arid and Saharan environment, and can live in sympatry with *A. algirus* [10,12].

The studied hedgehogs were infested by four tick taxa: *Haemaphysalis erinacei*, *Rhipicephalus sanguineus*, *Ixodes* spp., and *Hyalomma aegyptium*; and two flea species: *Archaeopsylla erinacei* and *Ctenocephalides felis*. Similar results have been reported in Algeria, where *Atelerix algirus* were infested by *Rh. sanguineus* and *Hae. erinacei* ticks and *Archaeopsylla erinacei* fleas [18,19,32]. In Europe, hedgehogs have been reported to be infested by *Ixodes ricinus*, *Ixodes hexagonus*, and *Archaeopsylla erinacei* [15,16,17]. In this study, we screened hedgehogs and their infesting arthropod vectors (ticks and fleas) for VBPs. In this context, a positive tick or flea meant it contained similar DNA sequences to targeted genes of VBPs, but this did not necessarily mean that the VBP was actually present in the arthropod [36]. Identification of pathogens’ DNA in these ectoparasites suggests its presence and circulation in the studied localities, but these arthropod species are not necessarily their biological vectors. However, it should be noted that in this survey, the main pathogens were most often identified in their natural vector as well as in their reservoir hosts, suggesting a stronger link between a pathogen, its natural vector, and the wild host than with other arthropod vector species.

### 3.1. Coxiella burnetii in Hedgehogs, Ticks and Fleas

Using microfluidic real-time PCR tests, we detected *Coxiella burnetii* DNA, the pathogenic agent of Q fever, in two *A. algirus* from Bizerte region. This mammal was positive in more than one biopsy, suggesting possible amplification of the pathogen in hedgehog organs. Our results are consistent with those of a recent study which reported the infection of *Erinaceus amurensis* hedgehogs in China by *C. burnetii* [30]. However, hedgehogs are known to carry the etiological agent of Q fever [37], since *C. burnetii* antibodies have been detected among 64 tested European hedgehogs in Austria [38]. Moreover, *C. burnetii* has also been described worldwide in domestic and wild animals, such as red foxes, rodents, and wild birds [39,40,41], while reservoirs are extensive but not accurately identified [42].

This study also provides the first detection of *C. burnetii* in hedgehog’s *Haemaphysalis erinacei* tick and *Archaeopsylla erinacei* flea. To our knowledge, this bacterium has never been detected in these tick and flea species; therefore, we may defend the hypothesis that hedgehog fleas and ticks may be vectors of *C. burnetii* among wild mammals. Data about flea infection with *C. burnetii* are scarce, but this pathogen has been reported in other flea species, such as *Ctenocephalides canis* and *C. felis*, and *Xenopsylla cheopis* infesting foxes and rats, respectively [43,44].

Likewise, in this study, we report *C. burnetii* DNA in *Rh. sanguineus*, the brown dog tick collected from hedgehogs. This tick species is known to harbor this pathogen [45,46]. *Coxiella burnetii* was also isolated from *Rh. sanguineus* infesting a dog naturally affected by Q fever [47]. In Algeria, a neighboring country to Tunisia, *C. burnetii* has recently been detected in *Ixodes vespertilionis* ticks infesting bats, but not in *Rh. sanguineus* infesting the *A. algirus* hedgehog [32]. Our data are worrisome in the context of a one health approach, as *Rh. sanguineus* is widespread and may also feed on a wide range of domestic and wild animals, as well as on humans [48]. In this context, we hypothesize that *A. algirus* hedgehogs in Tunisia could be an efficient wild reservoir for *C. burnetii* and could participate in its enzootic cycle, which involves ticks and fleas enhancing the exposure of domestic animals and humans to this zoonotic and pathogenic agent. In addition, *Coxiella burnetii* infection has recently been reported in dromedary camels in Tunisia [49], and the bacterium infests *Hyalomma dromedarii* and *Hy. impeltatum* ticks [50]. In Tunisia, clinical Q fever is rarely reported by physicians, yet the genome of this pathogen has been sequenced from the heart valve of a Tunisian patient with severe infective endocarditis [51]. However, given the highly conserved nature of the 16S rRNA genes among *Coxiella* species, these results should be further confirmed by targeting other genes, as *Coxiella* in ticks may be *Coxiella*-like organisms [52,53,54].

### 3.2. Rickettsia *spp*. in Hedgehogs and Haemaphysalis erinacei Ticks

*Rickettsia* DNA was detected in two *A. algirus* and their infesting *Hae. erinacei* ticks from the Bizerte and Kef regions. Thus, horizontal transmission of *Rickettsia* spp. between *Hae. erinacei* ticks and *A. algirus* hedgehogs may occur. Among these two positive hedgehogs, one was infected in three different organs. This means that possible amplification of the pathogen in hedgehog organs may occur. As a result, *A. algirus* hedgehogs may play an important role in the enzootic cycle of this bacterium and could be a reservoir. In the same context, hedgehogs in China constitute potential reservoirs of *R. sibirica sibirica* (the agent of Siberian tick typhus), and its DNA was detected in feeding ticks [55]. Our results on infection of *Hae. erinacei* ticks by *Rickettsia* spp. corroborate those reported in Algeria, where *Hae. erinacei* collected from *A. algirus* hedgehogs were shown to be infected with *Rickettsia* sp. and *R. heilongjiangensis* [19]. Moreover, other *Rickettsia* species, such as *Rickettsia sibirica* subsp. *mongolitimonae* and *Rickettsia raoultii*, were also reported in *Hae. erinacei* infesting hedgehogs in Turkey [56] and marbled polecats at the China–Kazakhstan border, respectively [57]. As *Hae. erinacei* is known to feed on humans [58,59], potential infection of people who are in close contact with hedgehogs by these infectious agents should be considered. 

### 3.3. Rickettsia massiliae in Rhipicephalus sanguineus Ticks

Our analysis revealed that 8 of 15 tested *Rh. sanguineus*, collected from *A. algirus*, were positive for *R. massiliae*. Accordingly, *R. massiliae* has previously been detected in *Rh. sanguineus* ticks collected from domestic animals (dogs, sheep, and goats) in Tunisia [60,61]. In accordance with our results, *R. massiliae* was also detected in *Rh. sanguineus* collected from *A. algirus* and *Erinaceus europaeus* in Algeria and France, respectively [19,62]. Additionally, *R. massiliae* was detected in *Rh. sanguineus* and *Rh. turanicus* collected from wild and domestic animals in France, Spain, and Greece [62,63,64,65,66]. Therefore, the hedgehog may play a reservoir role for this *Rickettsia* species that may infect humans. Although few human cases have been reported [67,68,69], the pathogen was isolated in one case in Italy [70].

### 3.4. Rickettsia massiliae in Fleas and Rickettsia lusitaniae in Rhipicephalus sanguineus

Interestingly, we report *R. massiliae* in one *A. erinacei* flea. To our knowledge, this is the first report of *R. massiliae* DNA in a flea, since it is known to be primarily transmitted by ticks. In the same context, microorganisms that are considered tick-borne pathogens have been reported in fleas, such as *Babesia microti* in *Orchopeas leucopus* fleas collected from *Peromyscus leucopus* [71]. Similarly, *Anaplasma phagocytophilum* DNA was identified in fleas collected from red foxes [72] and *Borrelia burgdorferi* has been identified in fleas feeding on small mammals [73,74]. However, our results do not imply that fleas may be vectors of *R. massiliae*, as detection of this pathogen’s DNA may correspond to remains reflecting the fact that the engorged fleas co-infested with ticks on the same hedgehog.

In fact, in our investigation, the flea infected with *R. massiliae* was collected on a negative animal and thus does not relate to the hedgehog’s infection. Therefore, we hypothesize that (i) this hedgehog may have an infection that was not detectable at the time of sampling; (ii) this positive flea may have infested another, infected, host before infesting the tested negative hedgehog; or (iii) this flea acquired *R. massiliae* by co-feeding with an infected tick. In our case, the latter hypothesis may be explained by the fact that *Hae. erinacei* and *Rh. sanguineus* ticks collected from the same hedgehog were found to be infected by *Rickettsia* spp., and we already reported in this study the infection of *Rh. sanguineus* collected from other hedgehogs with *R. massiliae*. Nevertheless, to our knowledge, co-feeding between ticks and fleas has not been described before; only co-feeding between infected and naïve ticks [75,76,77], as well as between infected and naïve fleas [78].

Our study also revealed for the first time in Tunisia the presence of *Rickettsia lusitaniae* in a specimen of *Rh. sanguineus*. Interestingly, the other *Rh. sanguineus* ticks collected on the same hedgehog were infected with *R. massiliae*, while this animal was *Rickettsia* spp.-negative. Thus, the origin of *R. lusitaniae* infection remains unknown, since no data are available about its occurrence in hard ticks, while it has only been associated with Argasidae ticks, such as *Ornithodoros erraticus* in Portugal [79], *Ornithodoros yumatensis* in Mexico [80], and *Argas vespertilionis* in China [81,82]. Moreover, in Tunisia, *O. erraticus* tick species [83], a confirmed competent vector of *R. lusitaniae*, was collected in habitats of small wild mammals (rodents), which suggested this tick species may also feed on hedgehogs. 

### 3.5. Rickettsia asembonensis in Fleas

The third *Rickettsia* species detected in 71 of 91 (78%) tested *Archaeopsylla erinacei* fleas was *Rickettsia asembonensis*. Our results corroborated the unique reports made in Germany and Portugal, where *R. asembonensis* was detected in *A. erinacei* fleas collected from *Erinaceus europaeus* hedgehogs [84,85]. In contrast, *Rickettsia felis*, the agent of flea-borne spotted fever, was the common pathogen in *A. erinacei* sampled from European and North African hedgehogs [19,32,62,86], as well as from cats, dogs [87], and foxes [63]. To our knowledge, this is the first report of the occurrence of *R. asembonensis* in the Maghreb region; however, in Egypt, this *Rickettsia* was described in *Echidnophaga gallinacea* fleas [43]. Moreover, *R. asembonensis* DNA has been detected in other flea species within the cosmopolitan Pulicidae family [88], such as *Ctenocephalides felis*, where it was isolated [89]. This pathogen has mostly been detected in *C. felis* fleas collected from dogs, cats, and humans in Mexico, Brazil, USA, India, Malaysia, Rwanda, and Kenya [90,91,92,93,94,95,96,97,98]. Furthermore, *R. asembonensis* DNA has been detected in *Rhipicephalus sanguineus* ticks in Brazil [99] and Malaysia [100]. In addition, *R. asembonensis* was detected in cynomolgus monkeys (*Macaca fascicularis*) in Malaysia [101] and in cats in Thailand [102]. Given the expansion of the host range of the hedgehog flea *A. erinacei*, it may contribute to the dissemination of *R. asembonensis* in different domestic and wild animal hosts and humans. Importantly, *A. erinacei* fleas have been reported feeding on humans causing purpura pulicosa [103,104]. Moreover, recent studies reported the association of *R. asembonensis* with human pathogenicity in Peru [105] and Malaysia [106]. 

### 3.6. Ehrlichia ewingii and Candidatus Ehrlichia shimanensis in Hedgehogs and Haemaphysalis erinacei Ticks

Interestingly, *Ehrlichia ewingii*, the etiologic agent of human granulocytic ehrlichiosis (HGE), was detected in *Hae. erinacei* ticks and in the blood of one *A. algirus*. This suggests that *A. algirus* may be a reservoir for this zoonotic microorganism, and horizontal transmission between this hedgehog and its infesting ticks may occur. *Ehrlichia ewingii* DNA has been reported in *Haemaphysalis flava* collected from hedgehogs in China [107]. In Tunisia, an *Ehrlichia* species closely related to *E. ewingii* were detected in *Ixodes ricinus* and *Hyalomma scupense* [108]. This may suggest active circulation of this pathogen in Tunisia. 

However, another genotype of *Ehrlichia* spp., closely related to *Candidatus* Ehrlichia shimanensis, was only detected in hedgehog tissue samples. This emergent *Ehrlichia* pathogen was first described in *Haemaphysalis* *longicornis* by Kawahara et al., in 2006 [109] as a novel *Ehrlichia* species phylogenetically close to three zoonotic *Ehrlichia* species: *E. muris*, *E. ewingii*, and *E. chaffeensis*. Recently, *Ca*. E. shimanensis was detected in ticks collected from cattle in Malaysia [110]. Genetic variants closely related to the *Ca*. E. shimanensis group were reported in *Haemaphysalis* ticks in Japan [111]. We suggest that *A. algirus* hedgehogs in Tunisia may be reservoir hosts for these emerging *Ehrlichia* species. However, given the highly conserved nature of the 16S rRNA genes, further investigations are required to attempt to genotype these genetic variants. 

### 3.7. Theileria youngi and Hepatozoon *sp*. in Hedgehogs

In this study, we reported for the first time *Theileria youngi* in hedgehogs in Northern Tunisia. Consistent with our results, in Saudi Arabia, four *T. youngi* haplotypes were detected in *Paraechinus aethiopicus* hedgehogs [112]. In addition, other *Theileria* species were reported in hedgehogs in China, such as *T. lunwenshuni* and *Theileria* sp. [113]. *Theileria* spp. have been reported to infect several wild and domestic animals as well as their infesting ticks [114,115,116,117,118]. Furthermore, several *Theileria* species, such as *T. annulata* and *T. lestoquardi*, were reported in Tunisia in small ruminants, cattle, horses, and ticks [119,120,121]. 

Additionally, *Hepatozoon* sp. DNA was detected in the liver of one hedgehog. To our knowledge, this is the first report of *Hepatozoon* spp. in hedgehogs. These pathogens have been detected in several wild and domestic animals [122,123,124,125]. In Tunisia, zoonotic *Hepatozoon canis* has been described in dogs [126]. 

### 3.8. Anaplasma *sp*. and Borrelia *sp.*

*Anaplasma* sp. DNA was detected in one *Rh. sanguineus* tick. It clustered with *A. phagocytophilum* detected in ticks and ruminants in China and Portugal [127,128], *Candidatus* Anaplasma boleense detected in ticks and mosquitoes in China [129,130], and uncharacterized *Anaplasma* species in ticks from China [129]. Given the highly conserved nature of the 16S rRNA genes, our results need further investigation, since in Tunisia, *Anaplasma* spp. have been detected in various domestic animals and ticks [131]. Moreover, *Erinaceus europaeus* hedgehogs have been reported to be a reservoir for *Anaplasma* spp. and *A. phagocytophilum* [25,26,132]. Recently, *Anaplasma marginale* was reported in long-eared hedgehogs, *Hemiechinus auritus*, and their *Rhipicephalus turanicus* ticks in Iran [133]. 

Here, we also report the presence of *Borrelia* DNA in one *Rh. sanguineus* collected from a non-*Borrelia* infected *A. algirus* hedgehog. The related sequence clustered with uncharacterized *Borrelia* species from Iran and Portugal [134], and with relapsing fever *Borrelia* species, such as *B. theileri* detected in *Rhipicephalus geigyi* from Mali [135] and *B. lonestari* detected in ticks collected from humans in the USA [136]. *Borrelia burgdorferi* s. l. genospecies have also been reported in *Erinaceus europaeus* and *E. roumanicus* hedgehogs and their infesting ticks [137,138]. In Tunisia, *Borrelia* species were also reported in ticks, cattle, ruminants, and camels [139,140]. 

### 3.9. Bartonella *spp*. in Ticks and Fleas

Finally, we reported *Bartonella* spp. DNA in *Hae. erinacei*, *Rh. sanguineus* ticks, and *A. erinacei* fleas. Zoonotic *Bartonella* species have been reported in Tunisia in fleas infesting domestic animals, stray dogs, camels, and patients [141,142,143,144]. In Algeria, *Bartonella* spp. were reported to infect *Atelerix algirus* and fleas infesting hedgehogs [31,145]. Consistent with our results, *Bartonella* species were detected in *A. erinacei* collected from the northern white-breasted hedgehog [146]. In addition, *Bartonella* spp. were reported to occur in wildlife and their infesting arthropods in Madagascar, China, Mexico, Brazil, Norway, and Thailand [147,148,149,150,151,152]. 

### 3.10. Co-Infections in Hedgehogs, Ticks, and Fleas 

In this study, we reported co-infections in *Atelerix algirus* hedgehogs, ticks, and fleas with at least two VBPs. These results shed light on the potential role of *Atelerix algirus* hedgehogs as simultaneous wild reservoir of zoonotic pathogens of medical and veterinarian interest. Similar observations on double (*Borrelia burgdorferi* and *Anaplasma phagocytophilum*) infections in ticks infesting hedgehogs were reported in *Ixodes ricinus* collected from *E. europaeus* hedgehogs in Germany [132]. In addition, *E. europaeus* hedgehogs were reported to be co-infected by several *Borrelia* species, such as *B. afzelii*, *B. bavariensis*, and *B. spielmani* [138].

## 4. Materials and Methods

### 4.1. Experimental Design

In this study, we screened hedgehogs, and their infesting ticks and fleas for VBP infection using a high-throughput microfluidic real-time PCR system. The experimental design is shown in Figure 8.

### 4.2. Study Location and Sample Collection 

A total of 20 hedgehogs were captured alive at night, in rural and suburban areas close to inhabited houses (Table 1). A total of 10 hedgehogs were captured during a monitoring study of sporadic cutaneous leishmaniasis in an endemic area in Northern Tunisia, in two localities (Zaafrane and, Abida) in the El Kef governorate. In addition, 10 hedgehogs were captured in three localities (Dahmani, Oued Souani, and Kalaat Senan) in the El Kef governorate (*n* = 3), five localities (Metline, El Garia, Bazina, Bni Atta, and Joumine,) in the Bizerte governorate (*n* = 6), and one locality in the Kasserine governorate (*n* = 1).

### 4.3. Sample Processing and DNA Extraction 

Captured animals were transferred to Institut Pasteur de Tunis. The corresponding genus and species of each captured specimen were determined based on external morphological criteria [10].

The animals were carefully examined for the presence of ticks and fleas. The ectoparasites vectors were collected from hedgehogs using fine forceps, and were identified to the species level using corresponding identification keys [58,153,154]. Specimens were then stored at −80 °C until DNA extraction. Hedgehogs were then euthanized and biopsies from the organs were taken, labeled and stored at −80 °C for further analyses. 

For DNA extraction, ticks and fleas were processed individually; each specimen was first washed in 70% ethanol, then rinsed twice in sterile distilled water, and well dried on sterile filter paper. Biopsies and arthropod samples were mechanically homogenized in 80 µL PBS using an Omni Bead Ruptor 24 (Omni International Inc., Kennesaw, GA, USA) and 2.8 mm ceramic beads for 3 cycles at 6 m/s for 1 min. Homogenized samples were immediately used for genomic DNA extraction using a DNeasy Blood & Tissue Kit (Qiagen, Hilden, Germany), following the manufacturer’s instructions. Quality and quantity of the extracted DNA were evaluated using a spectrophotometer (NanoDrop^®^, Germany). 

### 4.4. DNA Pre-Amplification

DNA was pre-amplified in order to increase the pathogenic load in the sample prior to microfluidic real-time PCR screening using the Fluidigm PreAmp Master Mix (Fluidigm, San Francisco, CA, USA), according to the manufacturer’s instructions. A mixture of pathogen-specific primers was prepared by pooling equal volumes of forward and reverse primers of each targeted pathogen at a final concentration of 200 nM each. The reactions were performed in 5 µL as a final volume, containing 1 µL Fluidigm PreAmp Master Mix, 1.25 µL pooled primers mix, 1.5 µL Milli-Q water, and 1.25 µL DNA. A negative control, containing water instead of DNA, was added to each reaction. Pre-amplification reactions were performed using the following cycling program; one step at 95 °C for 2 min, 14 cycles at 95 °C for 15 s, and 4 min at 60 °C. The obtained pre-amplifcation products were diluted 1:10 and stored at −20 °C until microfluidic real-time PCR testing.

### 4.5. High-Throughput Real-Time PCR Screening

High-throughput microfluidic real-time PCR amplifications were performed using the BioMark™ real-time PCR system (Fluidigm, San Francisco, CA, USA) and 48.48 dynamic arrays enabling up to 2304 individual reactions to be performed in one run. Primers and probes used in this study are summarized in Appendix A. Real-time PCRs were performed using 6-carboxyfluorescein (6-FAM) and black hole quencher (BHQ1)-labeled TaqMan probes with PerfeCTa^®^ qPCR ToughMix^®^, Low ROX™ (Quanta Biosciences, Gaithersburg, MD, USA) following Michelet et al., 2014 [155]. The cycling conditions were as follows: 2 min at 50 °C and 10 min at 95 °C, followed by 40 cycles of two-step amplification for 15 s at 95 °C and 1 min at 60 °C. 

Three controls were included in each dynamic array chip: a negative water control to exclude contamination, a DNA extraction control (primers and probes targeting the 16S rRNA gene of ticks), and an internal control to exclude PCR inhibitors (*Escherichia coli* DNA strain EDL933 with specific primers and probes targeting the *eae* gene) [156]. Acquired data were analyzed using the Fluidigm real-time PCR Analysis Software (Fluidigm, USA).

### 4.6. Validation of Results by Conventional PCR, Nested PCR, and Sequencing

Conventional PCR/nested PCR using primers (Table 4) targeting genes or regions different from those targeted by microfluidic real-time PCR was carried out to confirm the infection by a specific pathogen (Table 4). PCR products were then sequenced by Sanger sequencing (Biomnis-Eurofins Genomics, Lyon, France). Sequences were analyzed using BioEdit Software (Ibis Biosciences, Carlsbad, CA, USA). The BLAST program (http://www.ncbi.nlm.nih.gov/BLAST (accessed on 12 January 2021)) was used to compare and analyze the sequence data.

### 4.7. Phylogenetic Tree Construction

Sequence alignments were performed using Muscle Software [165]. Maximum likelihood trees were generated by 1000 bootstrap repetitions based on the General Time Reversible model for *Anaplasma* spp., *Ehrlichia* spp., *Borrelia* spp., and *Theileria/Hepatozoon* spp. trees and the Kimura 2 parameter model and Tamura–Nei model for *Rickettsia* spp. and *Rickettsia massiliae* trees respectively, with MEGAX software [166]. The initial trees for the heuristic search were obtained automatically by applying neighbor-joining and BioNJ algorithms to a matrix of pairwise distances estimated using the maximum composite likelihood (MCL) approach, and then selecting the topology with a superior log likelihood value. The trees were drawn to scale, with branch lengths measured in the number of substitutions per site [167]. The codon positions included were 1st + 2nd + 3rd + Noncoding. All positions containing gaps and missing data were eliminated. 

## 5. Conclusions

Hedgehogs are hosts of hematophagous arthropods and may be considered competent reservoirs for several arthropod-borne zoonotic pathogens in Tunisia. However, it will be necessary to confirm the circulation of the identified pathogens, as the *Atelerix algirus* hedgehog is a peri-urban dweller, and to determine their possible dissemination to other wildlife and their infesting arthropod vectors. Our results are of concern from a medical standpoint as several zoonotic pathogens were detected in hedgehogs and their infesting arthropod vectors in different localities. These results emphasize the need for accurate surveillance of hedgehogs and their ticks and fleas. This may help prevent possible exposure risks in humans. 

## Figures and Tables

**Figure 1 pathogens-10-00953-f001:**
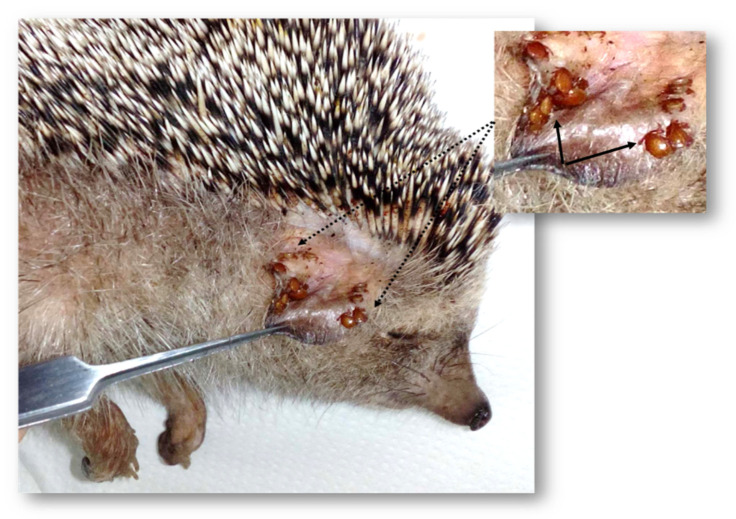
Photo of a hedgehog’s ear heavily infested by *Haemaphysalis erinacei* ticks. Full arrows shows higher magnification of the tick-infested region.

**Figure 2 pathogens-10-00953-f002:**
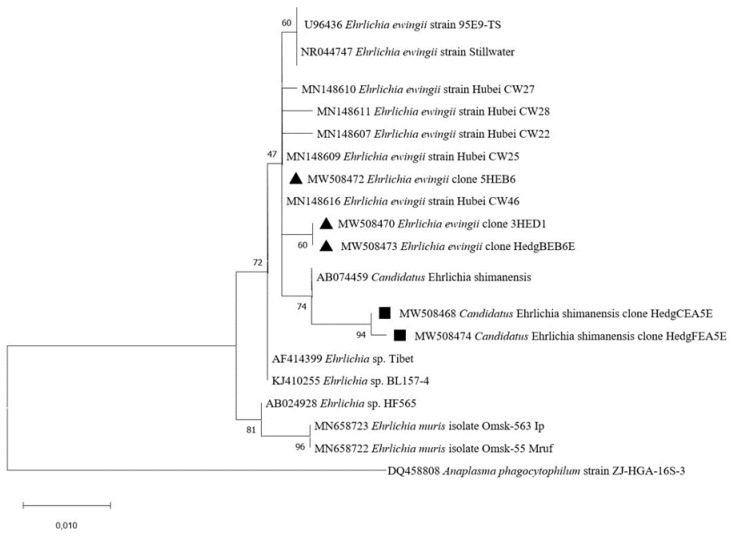
Phylogenetic analysis of 16S rRNA sequences of *Ehrlichia* spp. Phylogenetic analysis of 16S rRNA sequences of *Ehrlichia* spp. using the maximum likelihood method based on the General Time Reversible model. In the phylogenetic tree, GenBank sequences, species designations and strain names are given. The sequences investigated in the present study are marked with a black triangle (*E. ewingii*) and black square (*Ca*. E. shimenensis). The tree with the highest log likelihood (−952.31) is shown. The percentage of trees in which the associated taxa clustered together is shown next to the branches (bootstrap values). A discrete Gamma distribution was used to model evolutionary rate differences among sites (5 categories (+*G*, parameter = 200.0000)). The rate variation model allowed for some sites to be evolutionarily invariable ([+*I*], 37.52% sites). The tree is drawn to scale, with branch lengths measured in the number of substitutions per site. This analysis involved 22 nucleotide sequences. There were a total of 510 positions in the final dataset.

**Figure 3 pathogens-10-00953-f003:**
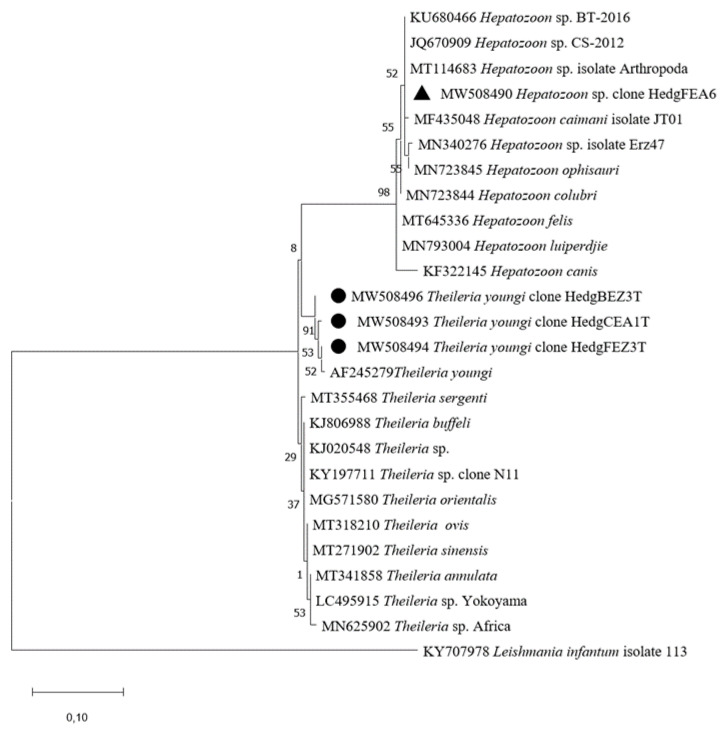
Phylogenetic analysis of 18S rRNA sequences of *Theileria* spp. and *Hepatozoon* spp. Phylogenetic analysis of 18S rRNA sequences of *Theileria* spp. and *Hepatozoon* spp. using the Maximum Likelihood method based on the General Time Reversible model. In the phylogenetic tree, GenBank sequences, species designations and strain names are given. The sequences investigated in the present study is marked with a black circle (*Theileria youngi*) and black triangle (*Hepatozoon* sp.). The tree with the highest log likelihood (−561.93) is shown. The percentage of trees in which the associated taxa clustered together is shown next to the branches (bootstrap values). A discrete Gamma distribution was used to model evolutionary rate differences among sites (5 categories (+*G*, parameter = 2.0666)). The tree is drawn to scale, with branch lengths measured in the number of substitutions per site. This analysis involved 26 nucleotide sequences. All positions containing gaps and missing data were eliminated (complete deletion option). There were a total of 250 positions in the final dataset.

**Figure 4 pathogens-10-00953-f004:**
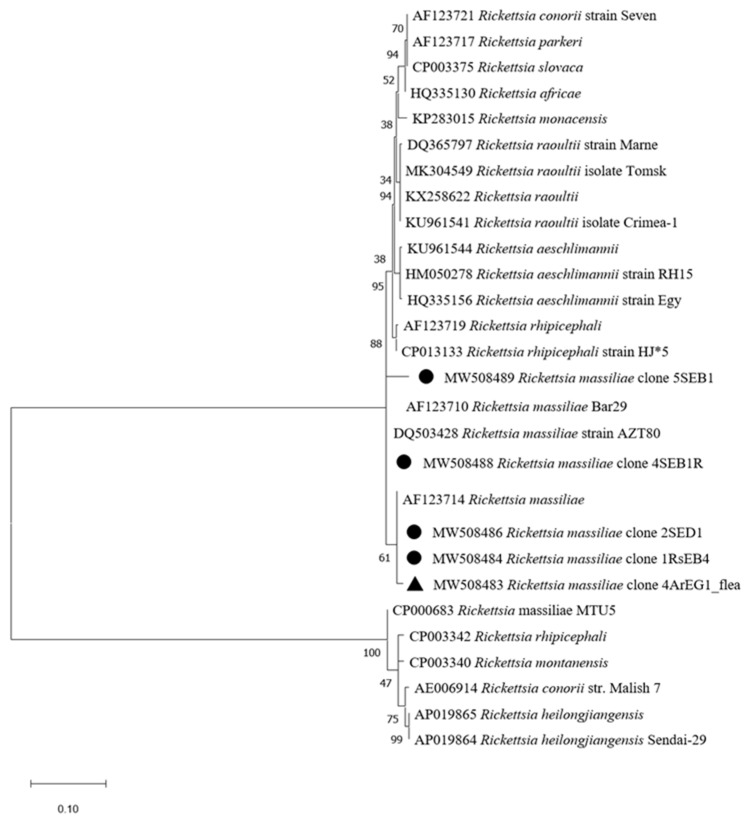
Phylogenetic analysis of ompB sequences of *Rickettsia* spp. Phylogenetic analysis of ompB sequences of *Rickettsia* spp using the maximum likelihood method based on the Tamura–Nei model. In the phylogenetic tree, GenBank sequences, species designations and strain names are given. The sequences investigated in the present study are marked with a black circle for the *R. massiliae* sequence recovered from *Rhipicephalus sanguineus* ticks and a black triangle for *R. massiliae* recovered from *Achaeopsylla erinacei* fleas. The tree with the highest log likelihood (−1768.76) is shown. The percentage of trees in which the associated taxa clustered together is shown next to the branches (bootstrap values). The rate variation model allowed for some sites to be evolutionarily invariable ([+*I*], 16.94% sites). The tree is drawn to scale, with branch lengths measured in the number of substitutions per site. This analysis involved 28 nucleotide sequences. There were a total of 662 positions in the final dataset.

**Figure 5 pathogens-10-00953-f005:**
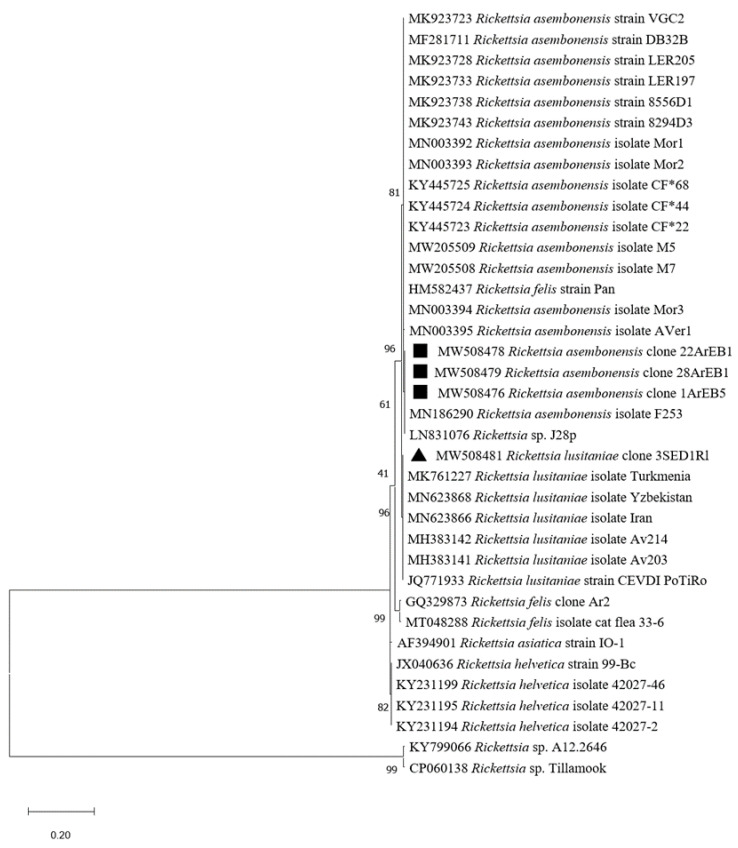
Phylogenetic analysis of gltA sequences of *Rickettsia* spp. Phylogenetic analysis of gltA sequences of *Rickettsia* spp. using the maximum likelihood method based on the Kimura 2-parameter model. In the phylogenetic tree, GenBank sequences, species designations and strain names are given. The sequences investigated in the present study is marked with a black square (*Rickettsia asembonensis*) and black triangle (*Rickettsia lusitaniae*). The tree with the highest log likelihood (−1134.16) is shown. The percentage of trees in which the associated taxa clustered together is shown next to the branches (bootstrap values). The rate variation model allowed for some sites to be evolutionarily invariable ([+*I*], 32.20% sites). The tree is drawn to scale, with branch lengths measured in the number of substitutions per site. This analysis involved 37 nucleotide sequences. There were a total of 410 positions in the final dataset.

**Figure 6 pathogens-10-00953-f006:**
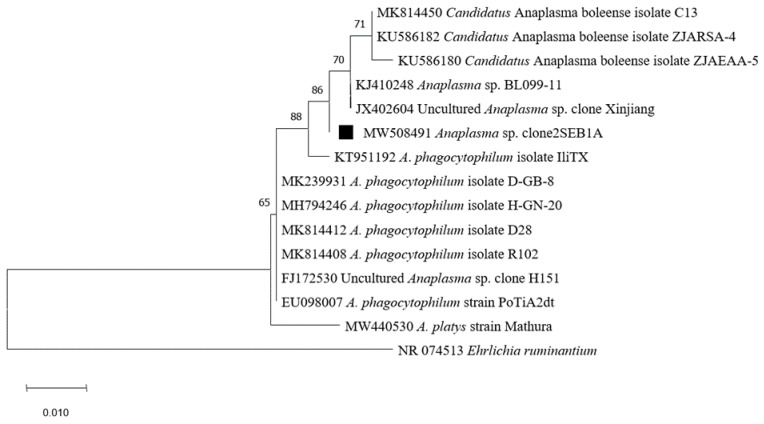
Phylogenetic analysis of 16S rRNA sequences of *Anaplasma* spp. using the maximum likelihood method based on the General Time Reversible model. In the phylogenetic tree, GenBank sequences, species designations and strain names are given. The sequence investigated in the present study is marked with a black square. The tree with the highest log likelihood (−851.75) is shown. The percentage of trees in which the associated taxa clustered together is shown next to the branches (bootstrap values). The rate variation model allowed for some sites to be evolutionarily invariable ([+*I*], 45.53% sites). The tree is drawn to scale, with branch lengths measured in the number of substitutions per site. This analysis involved 15 nucleotide sequences. There were a total of 459 positions in the final dataset.

**Figure 7 pathogens-10-00953-f007:**
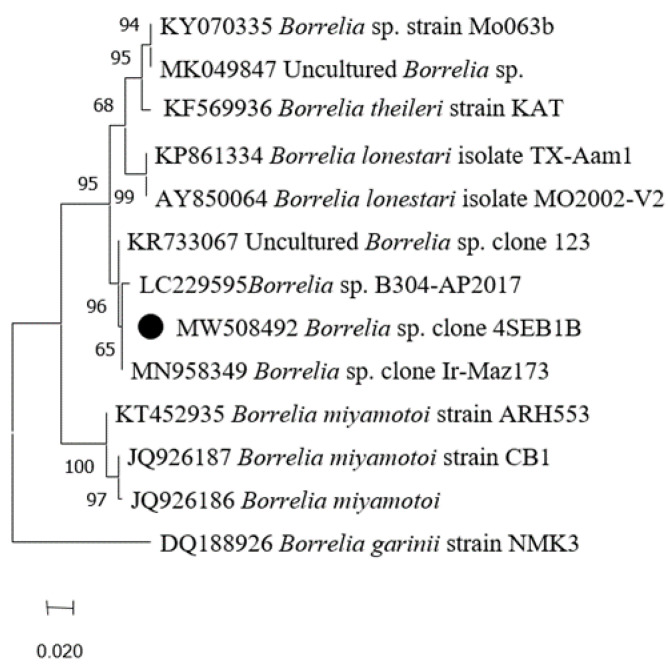
Phylogenetic analysis of flaB sequences of *Borrelia* spp. Phylogenetic analysis of flaB sequences of *Borrelia* spp. using the maximum likelihood method based on the General Time Reversible model. In the phylogenetic tree, GenBank sequences, species designations and strain names are given. The sequence investigated in the present study is marked with a black circle. The tree with the highest log likelihood (−679.69) is shown. The percentage of trees in which the associated taxa clustered together is shown next to the branches (bootstrap values). The rate variation model allowed for some sites to be evolutionarily invariable ([+*I*], 40.19% sites). The tree is drawn to scale, with branch lengths measured in the number of substitutions per site. This analysis involved 12 nucleotide sequences. There were a total of 265 positions in the final dataset.

**Figure 8 pathogens-10-00953-f008:**
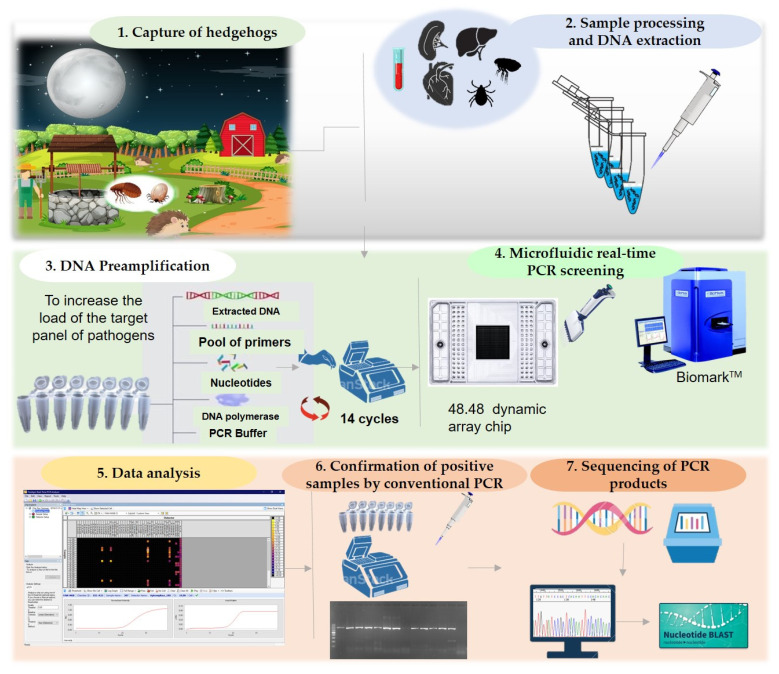
Schematic representation of the workflow used in this study.

**Table 1 pathogens-10-00953-t001:** Studied hedgehogs according to geographical location, sex, and ectoparasite infestation (ticks and fleas).

Hedgehog	Region	Locality	Geographical Coordinates	Sex ^a^	No of Collected Ticks ^b^	No of Collected Fleas ^b^
**ED1**	Kef	Dahmani	N: 35°56′35.606″ E: 8°49′50.747″	M	8	20
**EZ4**	Kef	Oued Souani	N: 36°11′49.6″ E: 8°58′33.964″	M	8	3
**EB1**	Bizerte	Metline	N: 37°14′56.022″ E: 10°02′29.616″	F	55	51
**EB2**	Bizerte	El Garia	N: 37°13′57.45″ E: 10°3′0.029″	F	1	0
**EB3**	Bizerte	Bazina	N: 36°57′49.392″ E: 9°18′0.158″	F	3	2
**EB4**	Bizerte	Bni Atta	N: 37°13′55.42″ E: 10°5′0.701″	M	8	4
**EB5**	Bizerte	Joumine	N: 36°55′34.248″ E: 9°23′14.744″	F	3	3
**EB6**	Bizerte	El Garia	N: 37°13′ 57.45″ E: 10°3′0.029″	M	7	5
**EG1**	Kasserine	Bouzguem	N:35°10′03″ E: 8°50′11″	M	17	4
**EA1**	Kef	Abida	N: 35°59.392″ E: 8°44′13.574″	F	Nd	Nd
**EA2**	Kef	Abida	N: 35°59.392″ E: 8°44′13.574″	F	Nd	Nd
**EA3**	Kef	Abida	N: 35°59.392″ E: 8°44′13.574″	F	Nd	Nd
**EA4**	Kef	Abida	N: 35°59.392″ E: 8°44′13.574″	F	Nd	Nd
**EA5**	Kef	Abida	N: 35°59.392″ E: 8°44′13.574″	M	Nd	Nd
**EA6**	Kef	Abida	N: 35°59.392″ E: 8°44′13.574″	M	Nd	Nd
**EA7**	Kef	Abida	N: 35°59.392″ E: 8°44′13.574″	M	Nd	Nd
**EZ1**	Kef	Zaafran	N: 33°26′39.271″ E: 8°55′18.39″	F	Nd	Nd
**EZ2**	Kef	Zaafran	N: 33°26′39.271″ E: 8°55′18.39″	M	Nd	Nd
**EZ3**	Kef	Zaafran	N: 33°26′39.271″ E: 8°55′18.39″	F	Nd	Nd
**ES1**	Kef	Kalaat Senan	N: 35°45′20.254″ E: 8°21′9.562″	M	0	0

^a^ F: female; M: male; ^b^ Nd: not determined.

**Table 2 pathogens-10-00953-t002:** Infection rates in hedgehogs, ticks, and fleas using microfluidic real-time PCR.

Pathogen	Hedgehog IR (Positive Samples/Total Tested Samples) ^a^	Tick IR (Positive Samples/Total Tested Samples) ^b^				Flea IR (Positive Samples/Total Tested Samples)	
		*Hae. erinacei*	*Rh. sanguineus*	*Ixodes* spp.	*Hy. aegyptium*	*Archaeopsylla erinacei*	*Ctenocephalides felis*
***Ehrlichia* spp.**	45% (9/20)	26% (24/92)	0	0	0	0	0
***Ehrlichia ewingii***	5% (1/20)	3.3% (3/92)	0	0	0	0	0
***Candidatus* E. shimanensis**	10% (2/20)	0	0	0	0	0	0
***Coxiella burnetii***	10% (2/20)	80.4% (74/92)	86.6% (13/15)	50% (1/2)	100% (1/1)	34% (31/91)	100% (1/1)
***Rickettsia* spp.**	10% (2/20)	40.2% (37/92)	86.6% (13/15)	50% (1/2)	0	82.4% (75/91)	100% (1/1)
***Rickettsia massiliae***	0	0	53.3% (8/15)	0	0	1.1% (1/91)	0
***Rickettsia lusitaniae***	0	0	6.7% (1/15)	0	0	0	0
***Rickettsia asembonensis***	0	0	0	0	0	78% (71/91)	0
***Bartonella* spp.**	0	3.3% (3/92)	6.7% (1/15)	0	0	2.2% (2/91)	0
***Theileria youngi***	40% (8/20)	0	0	0	0	0	0
***Hepatozoon* sp.**	5% (1/20)	0	0	0	0	0	0
***Borrelia* sp.**	0	0	6.7% (1/15)	0	0	0	0
***Anaplasma* sp.**	0	0	6.7% (1/15)	0	0	0	0

^a^ IR: infection rate; ^b^
*Hae*: *Haemaphysalis*; *Rh*: *Rhipicephalus*; *Hy*: *Hyalomma*.

**Table 3 pathogens-10-00953-t003:** Microfluidic real-time PCR results confirmed by sequencing of conventional PCR and nested PCR amplification products.

Pathogen Identification by Microfluidic Real-Time PCR	Host ^a^	Similarity %	Accession Number of the Reference Sequences	Pathogen Identification by Sequencing ^b^ (Targeted Gene)	Accession Number
***Anaplasma* spp.**	*Rh. sanguineus*	99.7	KJ410249	*Anaplasma sp*. (16S rRNA)	MW508491
***Anaplasma* spp.**	*Hae. Erinacei*	99.8	MN148616	*E. ewingii* (16S rRNA)	MW508469
***Ehrlichia* spp.**	*Hae. erinacei* *Atelerix algirus*	99.8–100	MN148616	*E. ewingii* (16S rRNA)	MW508471 MW508473
***Ehrlichia* spp.**	*Atelerix algirus*	99.3	AB074459	*Ca.* E. shimenensis (16S rRNA)	MW508474MW508475MW508468
***Rickettsia* spp.**	*A. erinacei*	100	MN186290 MK923741	*R. asembonensis* (gltA)	MW508476-MW508479
***Rickettsia* spp.**	*Rh. sanguineus*	100	MK761227	*R. lusitaniae* (gltA)	MW508481
***Rickettsia* spp.**	*A. erinacei*	99.3	AF123714	*R. massiliae* (ompB)	MW508483
***Rickettsia massiliae***	*Rh. sanguineus*	100	DQ503428	*R. massiliae* (ompB)	MW508482- MW508489
***Theileria* spp.**	*Atelerix algirus*	99	AF245279	*T. youngi* (18S rRNA)	MW508493- MW508496
***Hepatozoon* spp.**	*Atelerix algirus*	100	KU680466	*Hepatozoon* sp. (18S rRNA)	MW508490
***Coxiella burnetii***	*Atelerix algirus* *A. erinacei* *Hae. erinacei*	100	MN540441 LC46497	*C. burnetii* (16S rRNA)	MW508460- MW508464
***Borrelia* sp**.	*Rh. sanguineus*	99.7	MN958351	*Borrelia sp.* (flaB)	MW508492

^a^ *Rh: Rhipicephalus; Hae: Haemaphysalis; A: Archaeopsylla*; ^b^ *E: Ehrlichia;* Ca. E*: Candidatus* Ehrlichia*; R: Rickettsia; T: Theileria; C: Coxiella*.

**Table 4 pathogens-10-00953-t004:** Primers used in conventional PCR/nested PCR to confirm microfluidic real-time PCR results.

Pathogen	Targeted Gene	Primers	Sequence (5′-3′)	Amplicon Size (bp)	Reference
*Rickettsia* spp.	gltA	Rsfg877 Rsfg1258	GGGGGCCTGCTCACGGCGG ATTGCAAAAAGTACAGTGAACA	381	[157]
	ompB	Rc.rglt.4362p Rc.rompB.4836n	GTCAGCGTTACTTCTTCGATGC CCGTACTCCATCTTAGCATCAG	475	[158]
		Rc.rompB.4496p Rc.rompB.4762n	CCAATGGCAGGACTTAGCTACT AGGCTGGCTGATACACGGAGTAA	267	
*Anaplasma/Ehrlichia* spp.	16S rRNA	EHR1 EHR2 EHR3	GAACGAACGCTGGCGGCAAGC AGTA(T/C)CG(A/G)ACCAGATAGCCGC TGCATAGGAATCTACCTAGTAG	629	[159]
*Hepatozoon* spp.	18S rRNA	HepF HepR	ATACATGAGCAAAATCTCAAC CTTATTATTCCATGCTGCAG	660	[160]
*Theileria* spp.	18S rRNA	BABGF2 BABGR2	GYYTTGTAATTGGAATGATGG CCAAAGACTTTGATTTCTCTC	559	[161]
		BTH 18S 1st F BTH 18S 1st	GTGAAACTGCGAATGGCTCATTACR AAGTGATAAGGTTCACAAAACTTCCC	1500 bp	[162]
		BTH 18S 2nd F BTH 18S 2nd R	GGCTCATTACAACAGTTATAGTTTATTTG CGGTCCGAATAATTCACCGGAT	1500	
*Coxiella burnetii* and *Coxiella-like* endosymbionts	16S rRNA	Cox 16SF1 Cox 16SR1	CGTAGGAATCTACCTTRTAGWGG ACTYYCCAACAGCTAGTTCTCA	719–813	[52]
		Cox 16SF2 Cox 16SR2	TGAGAACTAGCTGTTGGRRAGT GCCTACCCGCTTCTGGTACAATT	625	
*Borrelia* spp.	flaB	FlaB280F FlaRL	GCAGTTCARTCAGGTAACGG GCAATCATAGCCATTGCAGATTGT	645	[163]
		FlaB737F FlaLL	GCATCAACTGTRGTTGTAACATTAACAGG ACATATTCAGATGCAGACAGAGGT	407	
*Bartonella* spp.	gltA	bart781 bart1137	GGG GAC CAG CTC ATG GTG G AAT GCA AAA AGA ACA GTA AAC A	380–400	[164]

## Data Availability

Not applicable.

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
