# Peer review of "Atelerix algirus, the North African Hedgehog: Suitable Wild Host for Infected Ticks and Fleas and Reservoir of Vector-Borne Pathogens in Tunisia"

_pathogens, 2021, doi:10.3390/pathogens10080953_

Round 1

Reviewer 1 Report

The submitted manuscript presents the results of an study whose main aim was to explore whether hedgehogs (Atelerix algirus) in Tunisia might contribute to the enzootic cycle of vector-borne bacteria (Rickettsia, Anaplasma, Ehrlichia, Bartonella, Borrelia, Coxiella, and Francisella) and protozoa (Babesia, Theileria, and Hepatozoon) by detecting the various pathogen`s DNA using a high-throughput microfluidic real-time PCR, both in the host hedgehogs tissues and their infesting tick and flea species.

This study appears to be the first large-scale high-throughput screening investigation of vector-borne bacteria and protozoa in Atelerix algirus hedgehogs and their ectoparasites to be conducted in Tunisia.

Overall, the manuscript is appropriately written, the analyses are adequately designed and the results obtained are well substantiated, indicating that the screening method utilised has a significant influencing factor in detecting the observed single and mixed infections.

There are several phylogenetic analises performed, and resulting trees are shown as figures, for the DNA sequences of all pathogens identified in this study except for the Coxiella DNA sequences, why is that?

The implication of hedgehogs in the transmission and maintenance of several emerging etiological agents of public health concern in Tunisia still needs to be investigated further, but no doubt the results presented in this investigation are a good start point to elucidated the posible participation of hedgehogs as a potential reservoir of vector-borne zoonotic pathogens.

Minor points

Table 1 Heading.

Could use “No.”; “N”; “Nr”; or “#” instead of “Nb” as abbreviation for number

Line 55 [10] [13]. Place reference number within the same square brackets

Line 64. Delete the period in “ [27]. And”…..

Line 442 Use “country” instead of county in …..”neighboring county to Tunisia”,

Line 451  Reference number “[5]” should be “[50]”?

Line 459 Correct “kef to “Kef”

Line 566 place “Hepatozoon canis” in italics

Line 574 [131,131]. Delete one reference number, or should it be [131,132].?

Line 695 “on the on the” delete repeat

Line 696 delete period in “model. for”

Line 741. Delete periods in abbreviated journal name

Line 780. Use abbreviatted form of journal name

Line 783. Delete periods in abbreviated journal name

Line 819. Use abbreviatted form of journal name “Parasit Vectors”

Line 828. Delete periods in abbreviated journal name

Line 911. Use abbreviatted form of journal name, such as in reference 65

Line 975. Delete periods in abbreviated journal name

Line 1035. Add initial of author’s first name “Gaowa,  ”

Line 1037. Delete periods in abbreviated journal name

Line 1039. Idem

Line 1130. Idem

Line 1134. Idem

Line 1161. Idem

Line 1182. Use abbreviatted form “Ticks Tick Borne Dis”

Line 1187. Delete periods in abbreviated journal name

Author Response

Responses to the reviewer 1 comments

Reviewer 1

Comment 1: There are several phylogenetic analyses performed, and resulting trees are shown as figures, for the DNA sequences of all pathogens identified in this study except for the Coxiella DNA sequences, why is that?

Response to Comment 1: Many thanks for the reviewer’s remark, we did not show the result of the phylogenetic analysis of Coxiella burnetii because we found that there was no significant genetic variability of the 16S rRNA targeted gene between the sequences we obtained and the Genbank reference sequences, we obtained only one cluster for all of them. Therefore, we did not put the phylogenetic tree of this pathogen as there will be no significant output.

Minor points

MP1: Table 1 Heading.

Could use “No.”; “N”; “Nr”; or “#” instead of “Nb” as abbreviation for number

Response to MP1: Thank you for this suggestion, we changed the Nb to No in the heading of the Table 1.

Line 55 [10] [13]. Place reference number within the same square brackets

Response Line 55: we did place the two references in the same brackets

Line 64. Delete the period in “ [27]. And”…..

Response Line 64: the period in “ [27]. And”….. was deleted

Line 442 Use “country” instead of county in …..”neighboring county to Tunisia”,

Response Line 442: Thank you for the precision we used country instead of county

Line 451 Reference number “[5]” should be “[50]”?

Response Line 451: Yes indeed, reference number 5 was replaced by 50 in the manuscript

Line 459 Correct “kef to “Kef”

Response Line 459: We did change kef to Kef

Line 566 place “Hepatozoon canis” in italics

Response Line 566: We replaced Hepatozoon canis to Hepatozoon canis

Line 574 [131,131]. Delete one reference number, or should it be [131,132].?

Response Line 574: We deleted one reference number, it is actually [131]

Line 695 “on the on the” delete repeat

Response Line 695: the repetition was deleted

Line 696 delete period in “model. for”

Response Line 696: the period in model. for was deleted

Line 741, Line 783, Line 828, Line 975, Line 1037, Line 1039, Line 1130, Line 1134, Line 1161 and Line 1187.. Delete periods in abbreviated journal name

Response Line 741, Line 783, Line 828, Line 975, Line 1037, Line 1039, Line 1130, Line 1134, Line 1161 and Line 1187: periods in abbreviated journal names were deleted in these Lines

Line 780, Line 819, Line 911, Line 1182. Use abbreviated form of journal name

Response Line 780, Line 819, Line 911 and Line 1182: abbreviated forms of journal names were added in these Lines

Line 1035. Add initial of author’s first name “Gaowa, ”

Response Line 1035: The initial of author’s first name Gaowa was added

Reviewer 2 Report

I have some comments on the attached file.

Figure legend is too long. On one figure, I give some comments on that.

Some result section have too much description about table and figures.

In some sentence, Spp. and Sp. is mixted.

Author Response

Responses to the reviewer 2 comments

Reviewer 2

Comment 1,Table 2 Heading: erinacei, start small letter

Response to comment 1, Table 2 Heading:  Hae. Erinacei was replaced by Hae. erinacei

Comment 2, Line 189: location of Figure legend. Figure expression range cannot identified

Response to comment 2, Line 189: the location of figure legend has been changed bellow the figure.

Comment 3; On the Table 2, it mentioned Theileria youngi not spp.

Response to comment 3 Line 203: We agree that Theileria spp. is not mentioned in the Table 2 so we deleted Table 2 reference from this sentence. Table 2 reference was added to line 213

Comment 4, Line 204: in  Line 174, there are same sentence, It is overlapped “Among these hedgehogs, three had two infected organs: liver and either heart, kidney or blood”

Response to comment 4, Line 204:

In the Line 174 we talk about Ehrlichia spp. infection in three hedgehogs (two different infected organs each)

In line 204, Theileria spp. infection  was also reported in other different three hedgehogs (two different infected organs each). These specimens are different from the ones reported in Line 174.

In the two Lines (174; 204), the same type of organs were reported to be infected in the six different hedgehogs

Comment 5, Line 208-209: I am not sure the data.

Response to comment 5, Line 208-209: In our microfluidic system we have primers and probes sets specific to the Theileria genus (Theileria spp.), T. velifera and T. mutans. The eight positive hedgehogs gave signals only with Theileria spp. sets. Three of these samples were randomly chosen and sequenced to confirm the Theileria infection species, thus all of them were revealed infected by Theileria youngi

Comment 6, Line 211: “The partial sequences of the 18S rRNA gene obtained (accession numbers MW508493, MW508494, and MW508496) were 98.9%-99.7% similar to Theileria youngi” I don't know how it was calculated

Response to comment 6, Line 212: In our study we obtained three sequences of Theileria youngi with some nucleotide differences to the reference sequences. The BLAST of these sequences using the online NCBI tool (https://blast.ncbi.nlm.nih.gov/Blast.cgi) revealed a percentage of identity of 98.9% and 99.7%.

Comment 7, Line 229-230, “Furthermore, the liver of one hedgehog (1/20, 5%) was positive for Hepatozoon spp. (Table 1)” On the Table 1, there is no data for pathogen.

Response to comment 7, Line 229-230: We agree with the Reviewer’s comment, Table 1 has been changed to Table 2

Comment 8, Line 235 “one hedgehog was infected in three different biopsies (spleen, liver, and lymph node), while the other hedgehog had a blood infection (Table 2)” There is no information about that on Table 2.

Response to comment 8, Line 235: We agree with the reviewer’s comment “Table 2” was deleted and added to the previous sentence.

Comment 9, Line 239: There is no phylogenetic analysis for Coxiella burnetii

Response to comment 9, Line 239: We did not show the result of the phylogenetic analysis of Coxiella burnetii because we found that there was no significant genetic variability of the 16S rRNA targeted gene between the sequences we obtained and the Genbank reference sequences, we obtained only one cluster for all of them. Therefore, we did not put the phylogenetic tree of this pathogen as there will be no significant output.

Comment 10, Line 243: There are minimum 4 samples (one hedgehogs, and spleen, heart, and liver sample). All samples did not amplified in PCR method. It is very weird situation.

Response to comment 10, Line 243: These samples needs further investigations and PCR optimization since their Ct values obtained by microfluidic real-time PCR system (and preamplification step) were low (23-28) corresponding to a Ct value of 35-40 in classical realtime PCR. Moreover depending of the type of samples, inhibitors could be present and avoid confirmation by PCR.

Comment 11, Line 251-252: “similar to C. burnetii strain CB-30 and C. burnetii strain SFA062 (accession numbers LC46497 and MN540441)”, I cannot find these results in Table 3.

Response to comment 11, Line 251-252: The second accession number of C. burnetii strain CB-30 (LC46497) was added in Table 3

Comment 12, Line 258: erinacei, start small letter

Response to comment 12 Line 258: Erinacei” was replaced by “erinacei”

Comment 13, Line “eight of 13 Rh. sanguineus ticks were found to be infected by Rickettsia massiliae (Table 2)”

Response to comment 13, Line 260: eight of 13 Rh. sanguineus ticks were found to be infected by Rickettsia massiliae was replaced byeight of 15 Rh. sanguineus ticks were found to be infected by Rickettsia massiliae”

Comment 14 Line 300-Line 306: move this paragraph near the Figure 2.

Response to Comment 14 Line 300-Line 306: The reviewer suggested to move the paragraph near figure 2. However, this paragraph describes the infection of only ticks by Ehrlichia species while the figure 2 describes the infection of hedgehogs and ticks by Ehrlichia species. Thus, the figure 2 was first placed near the hedgehogs infection paragraph.

Comment 15, Line 503-504: usually we harvested ectoparasite from wild rodent, we also collected ticks and fleas together. So it is possible

Response to comment 15, Line 503-504: we agree perfectly with your comment since we already mention in the same paragraph the following sentence: “(iii) this flea acquired R. massiliae by co-feeding with an infected tick”.

Comment 16 Line 560: Spp. not Sp.

Response to comment 16, Line 561: In this sentence, we mentioned uncharacterized Theileria species so we have to write sp. not spp. as described in the manuscript doi:10.1186/2049-9957-3-18

Comment 17, Line 569: Anaplasma Spp. not Sp.

Response to Comment 17, Line 569: In this sentence we could not identified the Anaplasma and Borrelia species so we assigned to this sequences Anaplasma sp. and Borrelia sp. as they mentioned sp. for uncharacterized species (doi:10.1186/s12862-014-0167-2).
